# Analysis of the Relationship between Stress Intensity and Coping Strategy and the Quality of Life of Nursing Students in Poland, Spain and Slovakia

**DOI:** 10.3390/ijerph17124536

**Published:** 2020-06-24

**Authors:** Ewa Kupcewicz, Elżbieta Grochans, Helena Kadučáková, Marzena Mikla, Marcin Jóźwik

**Affiliations:** 1Department of Nursing, Collegium Medicum, University of Warmia and Mazury in Olsztyn, 14C Zolnierska Street, 10-719 Olsztyn, Poland; 2Department of Nursing, Pomeranian Medical University in Szczecin, 48 Zolnierska Street, 71-210 Szczecin, Poland; grochans@pum.edu.pl; 3Department of Nursing, Faculty of Health, Catholic University in Ruzomberok, 48 A. Hlinku Street, 034-01 Ruzomberok, Slovakia; helena.kaducakova@ku.sk; 4Department of Nursing, University of Murcia, Campus de Espinardo, Edificio 23, 30100 Murcia, Spain; marmikla@yahoo.com; 5Murcian Institute of Biosanitary Research (IMIB), 30120 Murcia, Spain; 6Department of Gynecology and Obstetrics, Faculty of Medicine, Collegium Medicum, University of Warmia and Mazury in Olsztyn, 44 Niepodleglosci Street, 10-045 Olsztyn, Poland; marcin.jozwik@uwm.edu.pl

**Keywords:** stress, coping strategies, quality of life, student, nursing

## Abstract

*Background:* This study aimed to determine the relationship between stress intensity and coping strategies and the quality of life and health among nursing students in Poland, Spain and Slovakia. *Methods:* The study was performed on a group of 1002 nursing students from three European countries. A diagnostic survey was used as a research method and the data collection was based on the Perceived Stress Scale PSS-10, Mini-COPE Coping Inventory-and the WHOQoL-Bref questionnaire. *Results:* The average age of all the respondents was 21.6 years (±3.4). Most of the surveyed students rated their stress intensity over the last month as moderate or high. Comparison of the results of the stress levels in relation to the country of residence did not reveal statistically significant differences. In the group of Polish students, the most positive relationship between active coping strategies and the quality of life in the psychological (r = 0.43; *p* < 0.001) and physical health domain (r = 0.42; *p* < 0.001) were most strongly marked. Among Slovak students, significant correlations of low intensity were found between active coping strategies and the quality of life in the physical health (r = 0.15; *p* < 0.01), psychological (r = 0.21; *p* < 0.001), social relationships (r = 0.12; *p* < 0.05) and environment (r = 0.19; *p* < 0.001) domain. In overcoming stressful situations, Spanish students used the Sense of Humour strategy, which is considered less effective, although very useful in some cases. In this group, the strongest positive correlation was found for the psychological domain (r = 0.40; *p* < 0.001). *Conclusions:* There is a need to implement prevention and stress coping programmes at every stage of studies to ensure effective protection against the negative effects of stress and to improve the quality of life of nursing students.

## 1. Introduction

In the present world, the profession of a nurse has gained a new dimension and status. It is a modern and multi-tasking profession, requiring knowledge of nursing and medical sciences, social skills and competences. In the European Union Member States, basic nursing education is regulated by Directive 2005/36/EC of the European Parliament and of the Council of 7 September 2005 on the recognition of professional qualifications [1].

The training process in nursing programmes includes a theoretical and practical training component. The first contact of the nursing student with the practical dimension of the profession takes place during the clinical classes. Clinical education is that part of nursing education in which the student, as a member of a team and in direct contact with healthy or sick persons or communities of healthy or sick people, learns how, based on the acquired knowledge, skills and social competences, to organise, administer and evaluate the required comprehensive nursing care. The student not only learns teamwork but also the ability to lead a team and to organize general nursing care [1]. The functioning of students in contemporary university reality is related to the specific social roles they play. Academic youth, as a closed social group with a diverse structure, separated from the complex population of young people, encounter various obstacles and difficulties during their studies, often exceeding their ability to cope with a given situation [2]. Responses to stressful events and subjective evaluation of the events experienced may affect individual ways in which students deal with such events [3]. The literature points to three ways of understanding the concept of stress, namely: stress as a stimulus, stress as a response, and stress as a transaction. The above-mentioned ways of understanding the concept of stress are not considered mutually exclusive, but rather complementary [3].

In the first meaning the stress as a stimulus designates a situation inducing person’s tension and strong emotions. One of the theories supporting this definition is a concept of life changes proposed by Holmes and Rahe [4]. In the second understanding, stress generates physiological and psychological consequences as person’s reaction on stressing occurrence, as described in the concept of stress by Selye [5] and in the concept of homeostasis put forth by Cannon [6]. Finally, in the third meaning, the stress is explained as a relation of a person with their environment and is generated by an aggravating incident or an incident exceeding person’s capacity and threatening their well-being. Such an approach has been found in the transactional model presented by Lazarus and Folkman [7], as well as in the concept of resources by Hobfoll [8].

In recent years, persons’ activity aiming to deal with stressing events has been increasingly a study subject and is named “dealing with stress” [3]. There are different approaches of dealing with stress described in the literature. Referring to the transactional concept of stress, Lazarus and Folkman indicated two functions of dealing with stress: an instrumental one referred to dealing targeted on the problem, and a regulatory one bound to strategies concentrated on emotions [3].

As numerous studies have shown, nursing students face multiple new and difficult situations, which can be described as stressful events, during the clinical process of improving their professional skills, acquired during the programme-related courses [9,10,11,12]. Research suggests that there are three main sources of stress among students that are related to the academic environment, clinical practice and personal or social environment [13,14]. The intensity of the stress experienced in the clinical environment is primarily related to the care of the patient, accompanying another person in the face of death, performing nursing procedures and the burden of negative interactions with the staff and lecturers [15,16]. Spanish studies have confirmed that clinical stressors are perceived more intensively compared to academic and external stressors [17,18]. In other research conducted by Salvarani et al. in the group of 622 Italian students, it was demonstrated that “over 70% of nursing students reported a significant level of psychological stress” [19]. In turn, Bodys-Cupak et al. evaluated the patient-related stress factors and ways of coping with the stress of nursing students in southern Poland and found that students experienced moderate to high levels of stress, and as the level of stress increased, the frequency of coping with difficult situations through avoidance behaviour increased [20].

Particularly interesting results were obtained during the research conducted among nursing students from three countries: Greece, the Philippines and Nigeria. The results of the research indicate that stress intensity and the type of stressors and coping styles used by nursing students vary from country to country [21]. It has also been proven that higher levels of stress associated with clinical classes and unfavourable relationships with lecturers and the student community are associated with the sense of the quality of life of nursing students [22].

The issue of the quality of life is of particular importance in the context of the lives of academic youth. According to the definition of the World Health Organisation (WHO), quality of life is the subjective assessment by an individual of his or her life situation in relation to the culture in which he or she lives, his or her system of values, goals, expectations, interests [23]. An example of an extensive study on the quality of life, of international reach, is a project carried out by Cruz et al. aimed at identifying life situation of nursing students in Chile, Egypt, Greece, Hong Kong, India, Kenya, Oman, Saudi Arabia and the United States of America. The researchers have proven that the quality of students’ lives in the physical, psychological, environmental and social spheres varies and depends on many factors, including the age of the respondents, country of residence and monthly family income [24]. The subjective sense of well-being, on the other hand, was the subject of a study on the academic youth in Turkey. The aim of the survey conducted in the group of 396 nursing students, was to capture the relationship between life satisfaction and the quality of life evaluated in four domains [25]. In turn, Felicilda-Reynaldo et al. in international research on the quality of life of young nursing students, pointed out the importance of health in the spiritual dimension and the predictive roles of religiousness in assessing the students’ quality of life [26].

The research aimed at establishing a relationship between stress intensity, stress management strategies and the quality of life and health of nursing students in Poland, Spain and Slovakia.

## 2. Materials and Methods

### 2.1. Settings and Design

The research was conducted using the diagnostic survey method from May 2018 to April 2019. The participants of the study were students of the first degree–undergraduate, full-time studies in the programme of nursing at the University of Warmia and Mazury in Olsztyn, the Pomeranian Medical University in Szczecin (Poland), the University of Murcia in Murcia (Spain) and the Catholic University in Ružomberok (Slovakia). The criterion for inclusion in the study was the status of a nursing student, age up to 30 years and expressing consent to participate in the study. The criterion for exclusion from the study was the period of the examination session and absence of consent to participate in the study. The research was carried out at the place where the teaching classes for students are held. One of the researchers delivered the prepared sets of questionnaires to the universities where the research project was carried out. The participants in the study were introduced to the survey procedure and informed that the surveys are anonymous, voluntary and performed for the purpose of a research project. They were free to withdraw from the study at any time. After expressing their informed consent to participate in the study. The participants received sets of questionaries, then filled them personally out. The survey took on average about 20 min to complete. A total of 1017 survey forms were distributed among students. After collecting the data and eliminating defective questionnaires, 1002 (i.e., 98.5%) properly completed questionnaires were qualified for further analysis. The collected material was encoded in the Excel software and the results were analysed collectively.

### 2.2. Participants

The study involved 1002 students, including: 404 (40.3%) from Poland, 208 (20.8%) from Spain and 390 (38.9%) from Slovakia. The average age of all the respondents was 21.6 years (±3.4). The vast majority were women (91.32%). The age of the respondents was analysed in three age groups: ≤20 years, 21–22 years and ≥23 years Students participating in the study significantly differed in age (chi-square = 135.93; *p* < 0.001). Among Spanish students, 73.08% were 20 years old or under, half of the surveyed students from Slovakia were 21–22 years old, and students from Poland in the same age category accounted for a slightly smaller group (47.03%). There were 329 first-year students (32.83%), 458 s-year students (45.71%) and 215 third-year students (21.46%).

### 2.3. The Research Instruments

The research used the diagnostic survey method, and a self-constructed questionnaire was used to collect the studied variables, which included questions on sociodemographic data such as: place of residence (country), gender, age, level of education, form and year of study, and standardised research tools validated and available for general use in the mother tongue in each of the countries.

#### 2.3.1. PSS-10 Questionnaire

For estimation of stress intensity related to the life situation of students in the previous 30 days the Perceived Stress Scale PSS-10, developed by Cohen, Kamarck and Mermeldtein was used. The scale contains 10 questions about different subjective feelings related to personal problems and events, behaviours and ways of coping with them. Responses were given using the 5-point Likert scale. The respondent was to determine the extent to which he or she agreed with the statement by selecting appropriate values: 0—never, 1—almost never, 2—sometimes, 3—fairly often, 4—very often. According to the recommendations of the authors of the scale, before calculating the overall indicator of the perceived stress intensity, the scores in the answers to positively formulated questions, i.e., 4, 5, 7 and 8, were reversed, following the principle: 0 = 4; 1 = 3; 3 = 1; 4 = 0. The overall score of the scale is the sum of all points, with the theoretical distribution from 0 to 40. The higher the score, the higher the intensity of the perceived stress. In the original version, the internal reliability of the scale, assessed based on Cronbach’s alpha, ranges from 0.84 to 0.86 for the three samples tested by Cohen et al. [3,27,28,29].

#### 2.3.2. Mini-COPE Stress Management Inventory

Mini-COPE inventory developed by *C.S. Carver* is a self-report tool, used to measure coping dispositions, i.e., to assess typical responses and feelings in situations of intense stress. The psychometric properties of the original version of the Mini-COPE inventory are considered good (Cronbach’s alpha = 0.70) [3,30]. The inventory contains a total of 28 statements in 14 stress management strategies (two statements in each strategy). For each statement, the respondent marked one of the four possible answers: 0—I almost never do that, 1—I rarely do that, 2—I often do that, 3—I almost always do that. Each scale was evaluated separately, by adding the points for the answers concerning two statements making up the scale and dividing the sum by 2. The range of the results for each scale ranged from 0 to 3. The results were analysed separately for each strategy or grouped according to common features of the scale factor structure. The group of strategies defined as “Active Coping” includes the following strategies: Active Coping, Planning, Positive Revalidation. Another group of strategies referred to as “Helplessness” includes: Use of Psychoactive Substances, Cessation of Actions and Self-blame, while, “Seeking Support” includes the strategies: Seeking Emotional Support and Seeking Instrumental Support. “Avoidance Behaviours” include the following strategies: Taking Care of Something Else, Denial and Discharge. Such strategies as: “Turn to Religion”, “Acceptance” and “Sense of Humour” form independent groups [3].

#### 2.3.3. WHOQoL-Bref Questionnaire

The WHOQoL-Bref questionnaire is an abbreviated version of WHOQoL-100 developed by the quality of life researchers at WHO. WHOQoL-Bref is a generic questionnaire used to measure the overall quality of health and life. It helps to obtain a quality of life profile in terms of functioning in the physical health domain, the psychological domain, the social relationship domain and the environment domain. The WHOQoL-Bref questionnaire contains 26 questions. The respondents provided their answers based on the 5-point Likert scale (score range from 1 to 5, where 1 = very dissatisfied, 5 = very satisfied). In each domain, the respondent could obtain a maximum of 20 points. The results of individual domains are scaled in a positive direction (i.e., the higher is the score, the higher is the quality of life). The WHOQoL-Bref questionnaire additionally contains two questions defining the general health and the overall quality of life. The psychometric properties of the original version are considered very good (Cronbach’s alpha = 0.89) [31,32].

### 2.4. Statistical Analysis

The data generated during the *a posteriori* study were subjected to statistical analysis using the Polish version of STATISTICA 13 (TIBCO, Palo Alto, CA, USA). Socio-demographic data are presented as the number of cases and as the percentage values. The overall stress intensity ratio has been converted to standardized units, which were interpreted according to the characteristics of the sten scale. It contains 10 units and the scale jump equals 1 sten. Sten scores between 1 and 4 were considered low, between 5 and 6 were considered average and between 7 and 10 high [3]. The assessment of the significance of stress variation in the sten-scale was made using the chi-square test (χ^2^). The mean, standard deviation, confidence interval for mean value ±95%, median, minimum and maximum were used to describe the analysed variables. The ANOVA variance analysis test (F-test) comparing multiple samples of independent groups was used to investigate the significance of differences in the applied stress coping strategies, the PSS-10 stress indicator and WHOQoL-Bref quality of life indicators. To examine detailed significances between groups of students from Poland, Spain and Slovakia, a comparison of means for all ANOVA (F-test) groups was used. In turn, a Pearson (r) correlation was used to investigate the significance of the strength of the relationship between the PSS-10 stress indicator and the quality of life, and between the stress-coping strategy indicators and the quality of life. The strength of the relationship between the variables was interpreted based on Guilford’s classification [33]. For all of the tests, a significance level of *p* < 0.5 was assumed.

The presented research results are part of a larger international research project. The study meets the criteria for the cross-sectional study [34]. The study was conducted in line with the principles stated in the Declaration of Helsinki. The research project received a positive opinion (No. 4/2020) of the Senate Committee on Ethics of Scientific Research at the Olsztyn University.

## 3. Results

### 3.1. Differentiation of Stress Intensity (PSS-10) and Coping Strategies (Mini-COPE) Results among Students in Polish, Spanish and Slovak Research

An analysis of the data did not reveal statistically significant differences in stress intensity in students in terms of their country of origin (F = 0.03; *p* < 0.97).

The mean results of the observed stress of the Polish sample (18.60 ± 6.95) are similar to those of the Slovak (18.66 ± 4.70) and Spanish (18.55 ± 6.07) samples. After converting the overall stress ratio into standardised units, it was shown that a fairly large group of students from all the samples assessed the stress levels over the past month as high (Poland 49.8%; Spain 43.8%; Slovakia 45.4%). A small percentage (13.3%) of students from Slovakia and Spain (18.8%) achieved low scores, indicating low-stress levels related to their life situation (Figure 1).

The analysis of the data showed statistically significant differences in all stress coping strategies applied by students. In the group of strategies classified as active ways of coping with stress, manifested in taking action to improve the situation, Polish students revealed a greater intensity of the assessed remedial and adaptation strategies, i.e., Active Coping (F = 5.57; *p* < 0.004) and Planning (F = 87.37; *p* < 0.001) than students from other groups. On the other hand, students from Slovakia in a stressful situation chose the Positive Reframing strategy significantly more often than others (F = 3.95; *p* < 0.02).

In contrast, in the group of less effective strategies expressed as Helplessness, students from Spain obtained higher intensity of the strategies assessed, i.e., Use of Psychoactive Substances (F = 73.09; *p* < 0.001), Cessation of Actions (F = 45.92; *p* < 0.001) and Self-blame (F = 4.58; *p* < 0.01) than students from the other groups - these differences were statistically significant. It is worth noting the prevalence in the group of Spanish students of the Sense of Humour (F = 131.02; *p* < 0.001) and the Turn to Religion (F = 45.52; *p* < 0.001) strategies. Strategies forming a group defined as Seeking Support generally turn out to be adaptation strategies. The Emotional Support Seeking strategy concerns emotions and was significantly prevalent in the group of Polish students (F = 329.07; *p* < 0.001), while the Seeking of Emotional Support strategy, treated as a problem-focused strategy, was significantly more often applied in the group of Polish and Spanish students (F = 4.01; *p* < 0.02) than by Slovak students (Table 1).

### 3.2. Diversity of the Quality of Life (WHO-Bref) Results among Students in Polish, Spanish and Slovakian Research

A significant differentiation of the quality of life of nursing students is observed concerning the country of residence. Statistically significant differences of the quality of life of nursing students in terms of the country of residence were demonstrated in the perception of the overall quality of life (F = 12.22; *p* < 0.001) and general health quality (F = 14.63; *p* < 0.001) and in all four domains of life under analysis. The mean results of the general quality of life and general health of Spanish students are significantly lower (3.97 ± 2.22 vs. 3.65 ± 0.92) than Polish students (4.12 ± 0.7 vs. 4.02 ± 0.78) and Slovak students (4.11 ± 0.63 vs. 3.90 ± 0.76). Students from Poland obtained the highest average results in the social relationships domain (16.13 ± 2.89), similar to students from Slovakia (15.81 ± 2.94), while students from Spain attributed the highest value to the quality of life in the physical health domain (14.94 ± 2.23); (Table 2).

### 3.3. The Degree of Relationship between Stress Intensity and Quality of Life among Students in Polish, Spanish and Slovakian Research

The analysis of the data demonstrated a statistically significant relationship between stress intensity and the overall quality of life and general health and relationships between stress intensity and the four domains of the quality of life, i.e., physical health, psychological, social relationships and environment of nursing students. These are negative relationships, which means that the higher level of stress, the lower the quality of life and vice versa.

In the group of Polish students, significant negative relationships occur between the overall indicator of perceived stress and the overall quality of life (r = −0.48; *p* < 0.01) and the general health (r = −0.41); *p* < 0.001) at the average level, while this level is high for the physical health domain (r = −0.69; *p* < 0.01), psychological domain (r = −0.63; *p* < 0.01), social relationships (r = −0.44; *p* < 0.01) and environment domain (r = −0.57; *p* < 0.01).

Among Spanish students, stress intensity shows the strongest statistically significant negative relationship with the sense of the quality of life at a high level referring to the psychological (r = −0.59; *p* < 0.01) and physical health (r = −0.56; *p* < 0.001) domain, and at an average level with reference to the social relationships (r = −0.41; *p* < 0.001) and environment (r = −0.32; *p* < 0.001) domains. Correlation coefficients between stress intensity and the quality of life in the group of students from Slovakia took a negative direction and proved to be most significant at the average level in the psychological (r = −0.42; *p* < 0.001), environment (r = −0.42; *p* < 0.001) and physical health (r = −0.40; *p* < 0.001) domains (Figure 2).

### 3.4. The Degree of the Relationship between Stress Coping Strategies and the Quality of Life among Students in Polish, Spanish and Slovak Research

The analysis of the relationship between stress management strategies and the quality of life of nursing students showed that the most significant positive relationships were between active stress management strategies and all areas of quality of life for Polish students. However, the most important relationships between active stress management strategies and the quality of life in the psychological (r = 0.43; *p* < 0.001) and somatic (r = 0.42; *p* < 0.001) domain were the most strongly observed. These were positive relationships at the average level, which means that the more Polish students used active strategies involving less stress, the higher their quality of life. Weaker, but statistically significant, relationships of active stress management strategies with the quality of life in the physical health (r = 0.15; *p* < 0.01), psychological (r = 0.21; *p* < 0.001), social relationships (r = 0.12; *p* < 0.05) and environment (r = 0.19; *p* < 0.001) domains were also found in the group of Slovak students. Further analyses have revealed negative links between the evasive strategies expressed as helplessness and the quality of life of Polish students in the somatic domain (r = −0.43; *p* < 0.001) and psychological (r = −0.43; *p* < 0.001) at an average level. It is worth noting that in the group of Spanish students, none of the avoidance strategies, such as Cessation of Actions, Self-blame or Use of Psychoactive Substances, correlated with any quality of life components. The situation was also very similar in the group of Slovak students, only one negative correlation with the psychological domain was recorded (r = −0.16; *p* < 0.01). The strategies related to Seeking Emotional and Instrumental Support in the group of Spanish students did not show any correlation with the quality of life, but they revealed positive connections among Polish students in all dimensions of the quality of life and were most strongly marked in the domain of social relationships (r = 0.44; *p* < 0.001). Spanish students, on the other hand, use the Sense of Humour strategy in overcoming stressful situations, which is considered less effective, although very useful in some situations. The strongest relationship was recorded with the psychological domain (r = 0.40; *p* < 0.001); (Table 3).

## 4. Discussion

The results of this study conducted with the participation of nursing students in Poland, Spain and Slovakia show the complexity and variety of stressful experiences of students during their studies. As has been shown, the stress experienced and remedial activities undertaken by students modify their quality of life and health. In the current study, the average results of perceived stress on the PSS-10 scale for nursing students from Poland, Spain and Slovakia ranged from 18.55 ± 6.07 to 18.66 ± 4.70 and were slightly lower than in the cross-sectional research conducted at the University of Dammam in Saudi Arabia among nursing students (20.52 ± 7.59) [35]. Research conducted by various authors shows that the intensity of stress in other fields of study is at a similar level or significantly higher than in own research. An example is a study carried out in Barcelona at Autònoma University with a group of 4301 students who achieved average stress intensity results of very similar levels (20.29 ± 7.13) [36]. In turn, a Turkish study of 477 undergraduate students at a public university achieved a very high average stress intensity of 30.50 ± 6.10 [37]. A similar situation was observed in a French study which found that in a group of 483 students from the University of Paris Ouest Nanterre and other universities in the Paris region, the average stress intensity also amounted to 30.48 (±6.17) [38].

It should be emphasized that no statistically significant differences were observed in the current study as regards the measurement of student stress perception in relation to the country of residence. However, it was found that nearly half of the respondents (43.8% to 49.8%) perceived a relatively high-stress intensity, indicating quite intensive subjective feelings related to personal problems and events. A much more favourable situation among students was recorded in a study conducted by Albaqawi in Saudi Arabia. It was found that only 22.5% of the respondents experienced high levels of stress [35]. In turn, Labrague et al. also reviewed 11 items of research literature describing the perception of stress among nursing students from Saudi Arabia. The analysis of the results showed that students experienced and evaluated stress events in different ways, most often revealed moderate or high levels of stress, and applied both active and passive coping styles [39]. Other researchers also reached highly convergent conclusions. Reviewing 13 items of academic literature, Bhurtun et al. found that during their classes, nursing students assessed the events they encountered in terms of stress and experienced the negative effects of a stress transaction at a moderate or high level. The results indicated that academic teachers and nursing staff provided a strong stressor, since because students felt they were constantly being watched and evaluated [16]. In the current study, only 13.3% of students from Slovakia, 18.8% from Spain and 24.3% from Poland indicated low-stress levels over the previous month. Other Polish studies conducted by Bodys-Cupak et al. showed a low level of stress in 17.8% of the respondents, an average level in 26.7% of students, while 55.5% of the respondents indicated that they experienced intense stress [20].

The current study indicates that nursing students apply a variety of strategies to deal with stress. Statistically significant differences are related to the place of residence of students. Polish and Slovak studies demonstrate a fundamental similarity in the application of available (active) coping strategies, i.e., Active Coping, Planning, Positive Revalidation. The Spanish study, on the other hand, obtained significantly lower scores for active strategies and higher for avoidance strategies. Thus, the current study revealed an important role of individual differences in the course of coping processes. It can be concluded that a certain flexibility in dealing with stress increases the likelihood of adapting the strategy to the current situation related to studying and to cultural circumstances. The literature includes numerous studies indicating an important role of multiple factors directly or indirectly affecting stress coping effectiveness [36,37,38]. Important factors influencing the effectiveness of dealing with stress include the subjective properties of the individual, relating to the resources available to a given person. Interesting research on the positive role of disposable optimism, self-esteem and self-efficacy in dealing with stressful situations was conducted by Heinen et al. using a group of 321 medical students which found that the personal resources available to medical students acted as a buffer for the stress perceived [40]. In contrast, Alconero-Camarero et al. surveyed second-year nursing students at a Spanish university and found that emotional intelligence and styles of coping with stress were desirable characteristics of the students and played an important role in achieving satisfaction with their learning [41]. Fernández-Martínez et al. in their research also confirmed that emotional intelligence, a sense of coherence and coping with stress were positively related to the health of Spanish nursing students [42]. Li and Hasson as a result of their review of 12 literature items, synthesized evidence on the interaction of mental resilience, stress and the well-being of nursing students in different cultural environments. An analysis of the results showed that the level of mental resilience in students was moderate, while the stress level was high. The relationship between mental resilience and the stress and well-being of students was high [43]. A review of other studies showed that high levels of stress and anxiety in nursing students have an adverse effect on learning and increase fatigue [44]. There is evidence that high levels of stress can lead to serious health effects. This is confirmed, among others, by the results of research conducted in the United States. Researchers found that while studying in the nursing programme, students in many groups revealed much higher levels of anxiety symptoms and mental health problems than students in other programmes [45]. In order to minimize the effects of stress, numerous scientists researchers point to the need to identify the structure of stress factors causing tension and strong emotions in nursing students during their studies, with particular emphasis on clinical classes, and then to redesign, by the teachers, the didactic methods based on active and effective student-teacher cooperation [11,46].

In the current study, the intensity of stress and most of the coping strategies are related to the sense of the quality of health and life of nursing students. The results indicate that the quality of life is assessed differently by students in different countries. The highest score for the general quality of life and general health was obtained by students from Poland (4.12 ± 0.7 vs. 4.02 ± 0.78) and Slovakia (4.11 ± 0.63 vs. 3.90 ± 0.76). In another Polish study, conducted by Fidecki et al., nursing students rated the overall quality of life (3.98 ± 0.57) and overall health quality much lower (3.73 ± 0.74) [47]. In the current study, students from Poland and Slovakia most favourably evaluated the domain of social relationships, while students from Spain gave the highest marks to the physical health domain. The lowest marks in the Polish and Spanish group were given to the psychological domain, while in the Slovak group they were given to the physical health domain. Similarly, in Brazilian studies, the social relationships domain scored much higher than the physical health domain [48]. At the University of Hong Kong, a group of 538 nursing students also rated the social relationships domain the highest and the physical domain the lowest [49]. The physical health domain was rated the lowest in studies conducted by Aboshaiqah and Cruz in Saudi Arabia. The authors found that the quality of life of nursing students in the physical health, psychological, environment and social relationships spheres depended, among others, on the gender of the respondents, the year of study and monthly family income [50]. Norwegian studies showed a negative link between perceived stress and the quality of life of nursing students, while the sense of coherence proved to be a significant mitigation in the relationship between stress and the quality of life [51].

As the results of the current study indicate, the group of Polish students had the strongest positive relationship between active stress coping strategies and the quality of life in the psychological (r = 0.43; *p* < 0.001) and physical health (r = 0.42; *p* < 0.001) domains, while among Spanish students the strongest positive relationship was recorded between the Sense of Humour strategy and the psychological domain (r = 0.40; *p* < 0.001). Other studies conducted at three Spanish universities have shown that the dominant style of coping with stress among nursing students is the one focused on emotions [52]. This may indicate the dominance of the regulatory coping function, which helps nursing students to control the emotional response associated with a particular stressor. In contrast, students from Jordan, while attending clinical classes, most often used problem-solving strategies to cope with stress [53]. As can be seen in the Jordanian research, the instrumental coping function, which serves to control a stressor to reduce or remove its stressful properties, has been identified in the first place.

In summary, this study showed the differences in stress intensity, as well as different stress coping strategies depending on students’ life quality referred to studying. It may be inferred, that prophylactic and interventional measures should originate from recognized needs and problems of nursing students in each particular University, aiming to solve them directly.

## 5. Limitations and Implications for Practice

The results of the current study on the relationship between the general indicator of perceived stress, stress coping strategies and quality of life do not conclude the examination of this relationship. The authors of the study point to some limitations, related to the fact that the study did not exclude persons experiencing at the study time difficulties not related to studying nursing (i.e., family, financial, emotional problems not related to studies). Further research on a larger study group is desirable and systematic identification of stressors and/or severity of stressors in the academic community is postulated. There is a need to develop stress coping skills for nursing students by implementing permanent prevention and stress management programmes at every stage of their studies to ensure effective protection against the negative effects of stress and to improve the quality of life. Interventions involving psycho-educational workshops are recommended, which may involve assertive behaviour and conscious use of interpersonal communication techniques. Stress prevention programme should include attention training to improve concentration skills, reduce stress and increase resistance to negative emotions.

## 6. Conclusions

The conclusions of this study may be summarized in the following points:Among most of the surveyed nursing students, the intensity of perceived stress was moderate to high. No significant differences in stress intensity were observed in relation to the country of residence.Significant variations in all applied stress coping strategies were revealed. The majority of students use active ways of dealing with stress, which is manifested by a greater intensity of the assessed remedial and adaptation strategies.There are significant differences in the perception of the quality of life and health. Students from Poland and Slovakia gave the highest marks to the quality of life in the social relationships domain, while students from Spain gave the highest marks to the physical health domain.The quality of life and health of the surveyed students is significantly lower in people with high levels of stress.The degree of intensity of the relationship between the preferred coping strategies and the quality of life of the surveyed students varies. This mainly concerns situational coping in relation to specific events that occur during the course of study.

## Figures and Tables

**Figure 1 ijerph-17-04536-f001:**
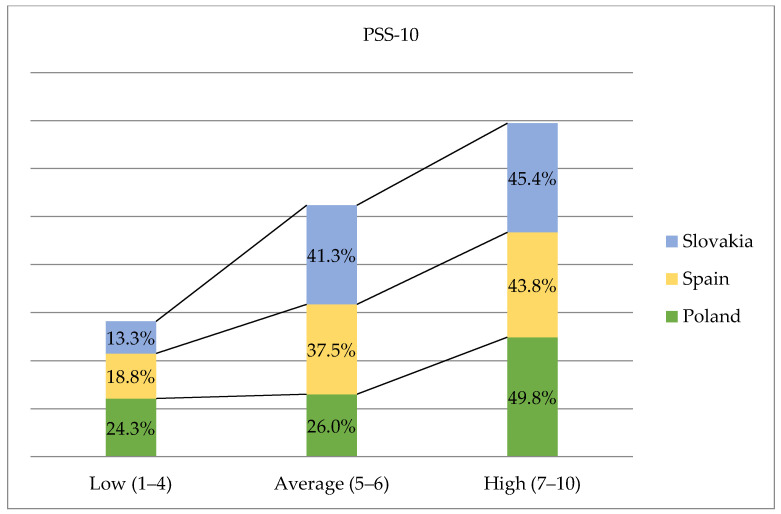
Percentage stress intensity results on the sten scale in Polish, Spanish, Slovakian studies.

**Figure 2 ijerph-17-04536-f002:**
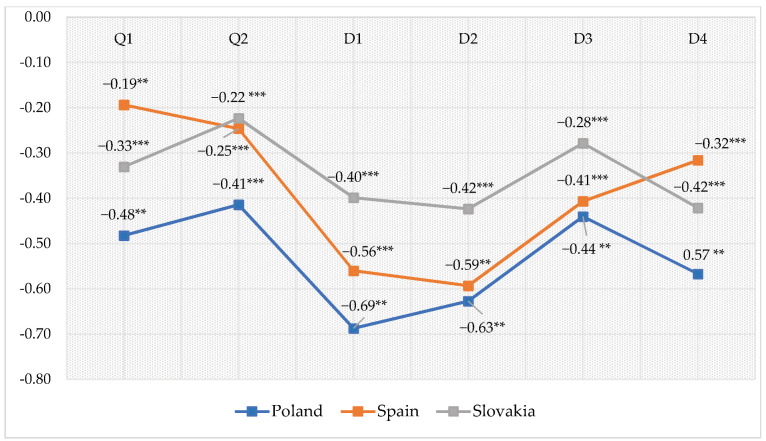
The character and the degree of the relationship between stress intensity and the quality of life in the surveyed students-Pearson’s correlation coefficients (r). Statistically significant: ** *p* < 0.01; *** *p* < 0.001. Explanation: Q1—satisfaction with the overall quality of life; Q2—satisfaction with the general health; D—physical health domain; D2—psychological domain; D3—social domain; D4—environment domain.

**Table 1 ijerph-17-04536-t001:** Stress intensity and coping strategies in the surveyed students-results of the significance test of differences including the grouping variable-country of residence.

Variables		Country of Residence			ANOVA (F-Test)	*p* Value
		Poland *n* = 404 (40.3%)	Spain *n* = 208 (20.8%)	Slovakia *n* = 390 (38.9%)		
		M ± SD, Me, Min.—Max., −95%, +95%	M ± SD, Me, Min.—Max., −95%, +95%	M ± SD, Me, Min.—Max., −95%, +95%		
Stress intensity (PSS-10)		18.60 ± 6.95, 19,	18.55 ± 6.07, 19,	18.66 ± 4.70, 19,	0.03	0.97
		0–35, 17.90–19.26	0–36, 17.72–19.38	0–33, 18.19–19.12		
Coping strategies groups acc. to Mini-COPE	Strategies for coping with stress					
1. Active coping	Active coping	2.11 ± 0.73, 2	1.85 ± 0.63, 2	1.98 ± 0.65, 2	5.57	*0.004*
		0–3, 2.04, 2.19	0–3, 1.86, 2.04	0–3, 1.92, 2.05		
	Planning	2.01 ± 0.73, 2	1.25 ± 0.56, 1.5	1.75 ± 0.65, 2	87.37	*0.001*
		0–3, 1.94, 2.08	0–3, 1.17, 1.33	0–3, 1.68, 1.81		
	Positive revalidation	1.70 ± 0.79, 2,	1.59 ± 0.65, 1.5,	1.77 ± 0.68, 2,	3.95	*0.02*
		0–3, 1.62, 1.77	0–3, 1.50, 1.68	0–3, 1.70, 1.84		
2. Helplessness	Use of psychoactive substances	0.59±0.82, 0	1.31±0.61, 1.5	0.60±0.75, 0	73.09	*0.001*
		0–3, 0.51, 0.67	0–3, 1.22, 1.39	0–3, 0.52, 0.67		
	Cessation of actions	0.76 ± 0.75, 0.5	1.23 ± 0.58, 1	1.18 ± 0.72, 1	45.92	*0.001*
		0–3, 0.69, 0.83	0–3, 1.15, 1.31	0–3, 1.10, 1.25		
	Self–blame	1.20 ± 0.86, 1	1.35 ± 0.56, 1.5	1.34 ± 0.73, 1.5	4.58	*0.01*
		0–3, 1.11, 1.28	0–3, 1.27, 1.43	0–3, 1.27, 1.41		
3. Seeking support	Seeking emotional support	2.07 ± 0.79, 2	0.57 ± 0.65, 0.5	1.97 ± 0.71, 2	329.07	*0.001*
		0–3, 2.00, 2.15	0–3, 0.48, 0.65	0–3, 1.90, 2.05		
	Seeking instrumental support	1.96 ± 0.78, 2	1.96 ± 0.64, 2	1.83 ± 0.72, 2	4.01	*0.02*
		0–3, 1.89, 2.04	0–3, 1.87, 2.05	0–3, 1.76, 1.90		
4. Avoidance behaviours	Taking care of something else	1.85 ± 0.75, 2	1.71 ± 0.65, 1.5	1.69 ± 0.74, 1.5	5.66	*0.004*
		0–3, 1.78, 1.92	0–3, 1.62, 1.80	0–3, 1.61, 1.76		
	Denial	0.86 ± 0.72, 1	1.96 ± 0.60, 2	1.23 ± 0.75, 1	163.38	*0.001*
		0–3, 0.79, 0.93	0–3, 1.87, 2.04	0–3, 1.16, 1.31		
	Discharge	1.53 ± 0.75, 1.5	1.98 ± 0.63, 2	1.44 ± 0.66, 1.5	43.62	*0.001*
		0–3, 1.45, 1.60	0–3, 1.90, 2.07	0–3, 1.38, 1.51		
5. Turn to religion		0.92 ± 0.96, 0.5	1.62 ± 0.68, 1.50	1.39 ± 1.02, 1	45.52	*0.001*
		0–3, 0.83, 1.01	0–3, 1.52, 1.71	0–3, 1.29, 1.49		
6. Acceptance		1.85 ± 0.72, 2	0.51 ± 0.65, 0.5	1.80 ± 0.65, 2	310.88	*0.001*
		0–3, 1.78, 1.92	0–3, 0.42, 0.60	0–3, 1.74, 1.86		
7. Sense of humour		1.27 ± 0.73, 1.5	2.03 ± 0.68, 2	0.96 ± 0.86, 1	131.02	*0.001*
		0–3, 1.20, 1.34	0–3, 1.93, 2.12	0–3, 0.87, 1.04		

Statistically significant: *p* < 0.05; *p* < 0.01; *p* < 0.001. Explanation: M—mean, SD—standard deviation, Me—median, Min.—minimum, Max.—maximum, confidence interval for the mean value ±95%.

**Table 2 ijerph-17-04536-t002:** Quality of life of the surveyed students - results of the significance test of differences taking into account the grouping variable - country of residence.

Variables		Country of Residence			ANOVA (F-Test)	*p* Value
		Poland *n* = 404 (40.3%)	Spain *n* = 208 (20.8%)	Slovakia *n* = 390 (38.9%)		
		M ± SD, Me, Min.—Max., −95%, +95%	M ± SD, Me, Min.—Max., −95%, +95%	M ± SD, Me, Min.—Max., −95%, +95%		
WHOQoL-Bref	Q1	4.12 ± 0.7, 4,	3.97 ± 2.22, 4,	4.11 ± 0.63, 4,	12.22	*0.001*
		0–5, 4.04–4.19	0–5, 3.67–4.27	0–5, 4.04–4.17		
	Q2	4.02 ± 0.78, 4,	3.65 ± 0.92, 4,	3.90 ± 0.76, 4,	14.63	*0.001*
		2–5, 3.95–4.10	1–5, 3.53–3.78	1–5, 3.83–3.98		
	D1	15.07 ± 2.69, 15.43,	14.94 ± 2.23, 14.86,	13.97 ± 2.01, 14.29,	24.29	*0.001*
		8.57–20.00, 14.81–15.34	6.86–20.00, 14.63–15.24	8.00–19.43, 13.77–14.17		
	D2	14.67 ± 2.16, 15.00,	14.09 ± 2.12, 14.00,	14.39 ± 1.81, 14.67,	5.75	*0.003*
		6.67–19.33, 14.45–14.88	8.00–18.67, 13.80–14.38	8.00–18.00, 14.21–14.57		
	D3	16.13 ± 2.89, 16.00,	14.58 ± 3.05, 14.67,	15.81 ± 2.94, 16.00,	19.53	*0.001*
		6.67–20.00, 15.85–16.41	6.67–20.00, 14.17–15.00	5.33–20.00, 15.52–16.10		
	D4	14.98 ± 2.33, 15.00,	14.42 ± 2.20, 14.5,	14.42 ± 2.29, 14.5.	7.20	*0.001*
		9–20.00, 14.75–15.21	7–19.5, 14.12–14.55	5.5–20.00, 14.19–14.65		

Statistically significant: *p* < 0.05; *p* < 0.01; *p* < 0.001. Explanation: Q1—satisfaction with the overall quality of life; Q2—satisfaction with the general quality of health; D1—physical health domain (activities of daily living, dependence on medicinal substances and medical aids, energy and fatigue, mobility, pain and discomfort, sleep and rest, work capacity) [31,32]; D2—psychological domain (body image and appearance, negative feelings, positive feelings, self-esteem, religion, spirituality, personal beliefs, thinking, learning, memory, concentration) [31,32]; D3—social relationships domain (personal relationships, social support, sexual activity) [31,32]; D4—environment domain (financial resources, freedom/physical safety and security, health and social care: accessibility and quality, home environment, opportunities for acquiring new information and skills, participation in and opportunities for recreation and leisure, physical environment (pollution, noise, traffic, climate and transport) [31,32]; M—mean, SD – standard deviation, Me—median, Min.—minimum, Max.—maximum, confidence interval for the mean value ± 95%.

**Table 3 ijerph-17-04536-t003:** Nature and degree of intensity of relationships between preferred stress management strategies and quality of life in the studied students-r-Pearson correlation coefficients.

Mini-COPE	Country	WHOQoL-Bref											
		Q1		Q2		D1		D2		D3		D4	
		r	*p*	r	*p*	r	*p*	r	*p*	r	*p*	r	*p*
1.	**Active Coping**												
	Poland	0.21	***	0.22	***	0.42	***	0.43	***	0.36	***	0.27	***
	Spain	0.06		0.03		0.04		0.17	*	0.03		0.05	
	Slovakia	0.08		0.01		0.15	**	0.21	***	0.12	*	0.19	***
2.	**Helplessness**												
	Poland	−0.24	***	−0.33	***	−0.43	***	−0.43	***	−0.30	***	−0.29	***
	Spain	0.08		0.09		0.04		0.09		−0.02		0.02	
	Slovakia	−0.09		0.03		-0.09		−0.16	**	−0.09		−0.04	
3.	**Seeking Support**												
	Poland	0.30	***	0.21	***	0.37	***	0.35	***	0.44	***	0.27	***
	Spain	−0.04		0.07		0.08		0.13		0.10		0.03	
	Slovakia	0.06		−0.02		0.10	*	0.05		0.08		0.16	**
4.	**Avoidance Behaviours**												
	Poland	−0.15	**	−0.10	*	−0.19	***	−0.17	***	−0.08		−0.15	**
	Spain	0.12		0.12		0.08		0.22	**	0.13		0.19	**
	Slovakia	−0.11	*	−0.01		0.01		−0.09		−0.06		−0.04	
5.	**Turn to Religion**												
	Poland	0.00		−0.06		−0.01		0.03		0.04		0.04	
	Spain	−0.10		−0.17	*	−0.15	*	−0.14	*	−0.21	**	0.05	
	Slovakia	0.13	*	0.07		0.07		0.13	**	0.00		0.12	*
6.	**Acceptance**												
	Poland	0.15	**	0.04		0.25	***	0.20	***	0.14	**	0.15	**
	Spain	0.15	*	−0.02		−0.06		0.03		−0.01		−0.05	
	Slovakia	0.13	**	−0.03		0.15	**	0.16	**	0.03		0.12	**
7.	**Sense of Humour**												
	Poland	0.01		−0.02		0.09		0.05		0.06		**	
	Spain	0.20	**	0.21	**	0.28	**	0.40	***	0.35	***	0.30	***
	Slovakia	−0.07		−0.08		−0.03		−0.05		−0.07		−0.06	

Statistically significant: * *p* < 0.05; ** *p* < 0.01; *** *p* < 0.001. Explanation: r-Pearson, Q1—satisfaction with the overall quality of life; Q2—satisfaction with the general health; D1—physical health domain; D2—psychological domain; D3—social domain; D4—environment domain.

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
