# Peer review of "Analysis of the Relationship between Stress Intensity and Coping Strategy and the Quality of Life of Nursing Students in Poland, Spain and Slovakia"

_ijerph, 2020, doi:10.3390/ijerph17124536_

Round 1

Reviewer 1 Report

Dear authors.

Thank you very much for the opportunity of reviewing this paper.

This is a very good job, however there are some aspects which should be take into account for publishing.

First of all, your introduction section is very brief. A deeper introduction of your variables and their relations is necessary. There are some interesting stress and coping models that would be interesting to explain. You mention coping strategies in your discussion that you did not introduce in the introduction section.

It would be interesting, in the data collection procedure, to explain the way that the participants filled the questionnaires (in person, via mail, email...). 

In the other hand, it would be interesting to know the alpha values of the instruments in your study.

Results and discussion sections are well constructed.

The overall paper is good, so I think that with these changes should be publishable.

Thank you very much.

Kind regards.

Reviewer 2 Report

There  should be noted,how the research tools were translanted in to the different countries languages.

Reviewer 3 Report

Stress levels of students enrolled in higher education, particularly professional education programs, is an area that all educators should be concerned about and aware of. Thank you for sharing your research into this issue in the context of nursing students from three countries. I found your manuscript enlightening and easy to read. I was particularly impressed with your response rate! I did have a few comments and questions regarding the manuscript that you might consider in terms of revising the manuscript to add some clarity and make it even more useful to readers:

  1. In your description of the PSS-10, it might be useful to point out that the questionnaire specifically asked respondents to evaluate the items based on the past 30 days, so that readers unfamiliar with the instrument have that added context.
  2. On line 212, you state "it was proven that a fairly large group of students 212 from all the samples assessed the stress levels over the past month as high". I might suggest writing "it was shown...", as there is some level of interpretation in defining the parameters for low, average, and high.
  3. It could be helpful to dive a bit more into the implications of your research in the discussion. For example:
    1. What considerations arise from the fact that some subgroups of students tend toward less effective or desirable coping methods?
    2. Should educator interventions be tailored to typical coping methods or lower-scoring areas in terms of quality of life?
    3. What specific considerations or interventions are suggested for nursing educators or for future research?
